# Estimation of Potassium Changes Following Potassium Supplements in Hypokalemic Critically Ill Adult Patients–A Patient Personalized Practical Treatment Formula

**DOI:** 10.3390/jcm10091986

**Published:** 2021-05-05

**Authors:** Amit Frenkel, Lior Hassan, Adi Segal, Adir Israeli, Yair Binyamin, Alexander Zlotnik, Victor Novack, Moti Klein

**Affiliations:** 1General Intensive Care Unit, Soroka University Medical Center, Beer-Sheva 8410101, Israel; motik@clalit.org.il; 2The Faculty of Health Sciences, Ben-Gurion University of the Negev, Beer-Sheva 8410101, Israel; yairben1@gmail.com (Y.B.); AleksZl@clalit.org.il (A.Z.); VictorNo@clalit.org.il (V.N.); 3Clinical Research Center, Soroka University Medical Center, Beer-Sheva 8410101, Israel; lior0351@gmail.com (L.H.); assaf.adi@gmail.com (A.S.); adirisraeli1@gmail.com (A.I.); 4The Joyce and Irving Goldman Medical School, Faculty of Health Sciences, Ben-Gurion University of the Negev, Beer-Sheva 8410101, Israel; 5Department of Anesthesiology, Soroka University Medical Center, Beer-Sheva 8410101, Israel; 6Anesthesia, Critical Care and Pain Medicine, Beth Israel Deaconess Medical Center, Harvard Medical School, Boston, MA 02215, USA

**Keywords:** intensive care unit, potassium, hypokalemia, formula

## Abstract

Hypokalemia is common among critically ill patients. Parenteral correction of hyperkalemia depends on dosages and patient characteristics. Our aims were to assess changes in potassium levels following parenteral administration, and to derive a formula for predicting rises in serum potassium based on patient characteristics. We conducted a population-based retrospective cohort study of adults hospitalized in a general intensive care unit for 24 h or more between December 2006 and December 2017, with hypokalemia. The primary exposures were absolute cumulative intravenous doses of 20, 40, 60 or 80 mEq potassium supplement. Adjusted linear mixed models were used to estimate changes in serum potassium. Of 683 patients, 422 had mild and 261 moderate hypokalemia (serum potassium 3.0–3.5 mEq/L and 2.5–2.99 mEq, respectively). Following doses of 20–80 mEq potassium, serum potassium levels rose by a mean 0.27 (±0.4) mEq/L and 0.45 (±0.54) mEq/L in patients with mild and moderate hypokalemia, respectively. Changes were associated with creatinine level, and the use of mechanical ventilation and vasopressors. Among critically ill patients with mild to moderate hypokalemia, increases in serum potassium after intravenous potassium supplement are influenced by several clinical parameters. We generated a formula to predict the expected rise in serum potassium based on clinical parameters.

## 1. Introduction

Hypokalemia is common among critically ill patients and may result from a wide spectrum of etiologies. Catecholamine levels are elevated among the majority of critically ill patients; this promotes potassium entry into cells via adrenergic receptors [1,2]. The subsequent increase in released insulin further shifts potassium into cells [1]. Moreover, alkalosis, which is frequent among ventilated patients, may contribute to this potassium shift, and consequently, to the risk of hypokalemia.

Correction of hypokalemia is crucial in critically ill patients and is usually achieved via the parenteral route. Standard doses via the intravenous route, in the range of 20–40 mEq, are generally used to correct hypokalemia in this context. The usual rate of potassium administration is 10 to 20 mEq/h in most patients, and up to 40 mEq/h for life-threatening hypokalemia [3]. Though most intensive care units follow similar protocols for potassium supplement, little is known about the associations between the following three crucial factors: the degree of hypokalemia, the supplied dose of potassium, and the expected increase in the serum potassium concentration [4,5].

The primary aim of the current study was to assess the increase in serum potassium concentration after potassium repletion, according to the degree of hypokalemia, in critically ill patients. The secondary aim was to derive a practical formula, based on a patient’s personal data, for calculating potassium deficit and predicting the expected rise in serum potassium after a given potassium supplement.

## 2. Methods

### 2.1. Study Population

We conducted a population-based retrospective cohort study at Soroka University Medical Center, a tertiary care medical center that serves as the only regional hospital in southern Israel (Beer-Sheva vicinity, estimated population of 100,000). We included all the patients aged 18 years or older who were hospitalized between December 2006 and December 2017 in the general intensive care unit (ICU) for 24 h or more with hypokalemia, with a serum potassium level equal to or greater than 2.5 mEq/L and less than 3.5 mEq/L. Accordingly, patients with severe hypokalemia (serum potassium less than 2.5 mEq/L) were not included.

### 2.2. Primary Exposure and Outcome Assessment

All the patients were treated with intravenous potassium supplement (a solution of potassium chloride) via a central line, in repeated doses of 20 mEq/L or 40 mEq/L. Serum potassium level was measured routinely by samples taken via the arterial line. For the purpose of this study, the patients were classified according to hypokalemia severity at their arrival at the ICU: mild hypokalemia (serum potassium 3–3.5 mEq/L) and moderate hypokalemia (serum potassium 2.5–2.99 mEq/L). To statistically cancel the influence of pH on serum potassium levels, all serum potassium levels were adjusted according to their pH scale, using the following formula:(1)$$K^{+}\mathrm{corr}=K^{+}- [\;0.6\;x\;\frac{\left(\mathrm{initial}\;\mathrm{pH}-\mathrm{final}\;\mathrm{pH}\right)}{0.1}]$$
K+ = Potassium (S, mEq/L); pH, initial (B, pH units); pH, final (B, pH units); * K+ increases 0.6 mmol/L for each 0.1 pH units decrease (and vice versa). B= blood, S= serum, P= plasma, U= urine.

The primary exposure was the potassium dosage, calculated as the sum of the total dosage of intravenous treatment given to each patient between two sequential potassium blood measurements. Patients were treated with potassium supplement in cumulative doses of 20 mEq, 40 mEq, 60 mEq and 80 mEq. The primary outcome was the delta of the extracellular potassium level, defined as the difference before and after treatment in serum potassium levels.

### 2.3. Statistical Analysis

Descriptive statistics are presented using summary tables. Continuous variables include means and standard deviations. Categorical variables are described with numbers and percentages. Comparisons between groups are presented by 95% confidence intervals and/or *p*-values. Percentages are rounded to one decimal place.

The method of analyses for continuous variables was the t-test for normally distributed variables, and non-parametric analysis using the Kruskal–Wallis test for variables not normally distributed. Categorical variables were tested using Pearson’s χ^2^ test for contingency tables or Fisher’s exact test, as appropriate.

In multivariable modeling, variables were selected according to clinical and statistical significance: First, baseline clinical characteristics and age; then, lab results.

For the dose response analysis, we used a linear mixed regression model. Patients’ personal identity numbers were used as random effects due to the repeated measurements nature of the data. Variables that were statistically significant (*p* < 0.1) in the univariable analysis were introduced to the model as fixed effects. Finally, we performed a Bland–Altman analysis to evaluate the reliability of the final model, by plotting the difference between the predicted and observed potassium levels vs. the average of the two measurements following potassium administration. All the analyses were performed using RStudio, version 1.1.423 (https://www.rstudio.com/). *p*-value of less than 0.05 was considered to indicate statistical significance.

## 3. Results

### 3.1. Study Population

The study comprised 683 patients: 422 with mild hypokalemia and 261 with moderate hypokalemia at admission. Table 1 summarizes characteristics of the study population. The majority of patients with mild and moderate hypokalemia were males (52.1% and 57.9%, respectively). Hypertension was the most common comorbidity among patients with mild hypokalemia, presenting in 7.6% (compared to 5.4% among those with moderate hypokalemia, *p* < 0.261). Diabetes mellitus was the most common comorbidity among patients with moderate hypokalemia, presenting in 6.1% (compared to 5.2% among those with mild hypokalemia, *p* < 0.611). Mortality was 23.5% and 21.5% for patients with mild and moderate hypokalemia, respectively (*p* = 0.544).

### 3.2. Characteristics of the Hospitalization Course–Table 2

For patients with moderate compared to mild hypokalemia, ICU duration and mechanical ventilation duration during ICU stay were not statistically different: 13.5 vs. 11.9 days, *p* = 0.152 and 7.4 vs. 6.7 days, *p* = 0.199, respectively. Vasopressor use during the ICU stay was also similar for the two groups: 13.8% and 11.1%, respectively (*p* < 0.302).

### 3.3. Serum Potassium Levels

Table 3 summarizes potassium levels among the patients with mild and moderate hypokalemia, before and after treatment with potassium chloride, according to four cumulative doses of intravenous supplements: 20 mEq, 40 mEq, 60 mEq, and 80 mEq.

Among the patients with *mild* hypokalemia at admission, the mean increase in serum potassium level was proportional to the increase in the supplementary doses, up to a dose of 60 mEq. Accordingly, the increase was 0.2 mEq/L for a dose of 20 mEq, 0.3 mEq/L for a dose of 40 mEq, and 0.4 mEq/L for a dose of 60 mEq (*p* = 0.012). However, such trend was not observed for the 80 mEq dose, for which the potassium level increased by only 0.3 mEq/L, similar to the increase observed following the 40 mEq potassium supplement dose (*p* = 0.42).

Among the patients with *moderate* hypokalemia at admission, the mean increase in serum potassium was 0.4 mEq/L following supplementation with 20 mEq dose; identical increases of 0.5 mEq/L were observed following supplementation with 40 mEq and 60 mEq doses. The increase dropped to 0.4 mEq/L following the 80 mEq dose.

The table also summarizes the distribution of cumulated potassium doses that were given between measurements of potassium levels. Most patients with mild and moderate hypokalemia were treated with cumulated doses of 40 mEq (65.6% and 74.2%, respectively). A single dose of 20 mEq between potassium level measurements was given to 22.8% of patients with mild hypokalemia, and to only 11% of those with moderate hypokalemia.

### 3.4. Linear Mixed Model Results

Table 4 depicts the results of the multivariable analysis of the adjusted potassium dose effect on the increase in serum concentration. A 20 mEq potassium chloride dose served as the reference dose for the three other doses (40 mEq, 60 mEq, and 80 mEq). Thus, for example, among patients treated with a dose of 40 mEq potassium chloride, the mean serum potassium increase was 0.15 mEq/L higher than that of those treated with a dose of 20 mEq, regardless of the initial potassium level before treatment. We adjusted the observed change, according to the initial creatinine levels: For every increase of 0.1 mg/dL in creatinine level, the serum potassium level increased by 0.03 mEq/L. Mechanical ventilation (B = 0.02, 95% CI = −0.05–0.09) and the use of vasopressors (B = 0.08, 95%CI = −0.01–0.16) were also associated with increased serum potassium level, while age was not. Lastly, the model showed a greater mean increase in serum potassium level after treatment, in patients with moderate compared to mild hypokalemia, by 0.2 mEq/L (*p* < 0.001).

### 3.5. Model Equation

Based on the findings of our study, we generated a model for estimating an effect of a given potassium dose, based on the following parameters: patient’s age, serum creatinine, the need for mechanical ventilation, and the use of vasopressors.
(2)$$Y\left(\mathrm{Potassium}\;\mathrm{delta}\right)=\left(\beta 0+\;\beta 1\times \mathrm{IV}\;\mathrm{potassium}\right)-\left(\mathrm{Dose}\;\mathrm{of}\;40+\;\beta 2\times \mathrm{IV}\;\mathrm{potassium}\right)\;-\left(\mathrm{Dose}\;\mathrm{of}\;60+\;\beta 3\times \;\mathrm{IV}\;\mathrm{potassium}\right)-\left(\mathrm{Dose}\;\mathrm{of}\;80+\beta 4\times \;\mathrm{Creatinine}\right)+\;\left(\beta 5\times \;\mathrm{Mechanical}\;\mathrm{ventilation}\right)+\left(\beta 6\times \;\mathrm{Vassopressor}\right)+\left(\beta 7\times \mathrm{Age}\right)+\left(\beta 8\times \mathrm{Moderate}\;\mathrm{hypokalemia}\right)$$

Figure 1 presents the results of the Bland–Altman plot of our model, with 95% CI. Most of the observations are within the 95% CI, indicating the reliability of the model in estimating post-treatment change in serum potassium levels. Although the plot distribution indicates no systemic bias, we emphasize that the model should be used cautiously in clinical practice due to the wide 95% CI. Analysis plot.

## 4. Discussion

A main finding of our study is the high degree of variability in the yield between the patients, even after adjustment for baseline clinical and demographic characteristics, and clinical factors. A second main finding is that the change in serum potassium level after each potassium dose can be predicted, using a model that takes into account patients’ clinical characteristics and laboratory data.

To our knowledge, only a few studies, none of them recent, investigated the association between the degree of hypokalemia, and the rise in serum potassium concentration after various doses of potassium supplements. In a study published in 1990, Kruse et al. [6] examined changes in potassium level after administration of potassium chloride infusions of 20 mEq, to 495 critical care patients. The mean potassium level before treatment was 3.2 mEq/L. The mean increment in serum potassium level per 20 mEq infusion was 0.25 mEq/L, which is similar to our current findings among patients with mild hypokalemia. However, that study did not analyze the data according to baseline potassium levels nor according to potassium treatment doses. In a study published one year later, Hamill et al. [7] examined 48 critically ill adult patients, according to three levels of hypokalemia. Each group was treated with a different potassium repletion dose according to the level of hypokalemia: 20 mEq, 30 mEq, and 40 mEq doses were given to patients with mild, moderate, and severe hypokalemia, respectively. The respective mean increases in potassium level were 0.5 ± 0.3 mEq/L, 0.9 ± 0.4 mEq/L, and 1.1 ± 0.4 mEq/L. These findings cannot be compared to our results, as we did not include patients with severe hypokalemia, and none of our patients received 30 mEq doses.

Our mixed model results showed several variables that were statistically associated with greater increases in serum potassium levels, among them renal failure (assessed by elevated creatinine level), the use of mechanical ventilation, and the use of vasopressors. The association of vasopressor use with a greater increase in serum potassium levels was unexpected. This is because, physiologically, increased levels of catecholamines act to promote potassium entry into cells via adrenergic receptors [1,2], and thus would be expected to induce hypokalemia. We do not have a definitive explanation for our findings. However, we assume that the use of vasopressors indicates sicker patients, who probably have higher cytokine levels. Notably, levels of the cytokine interleukin-6 were found to be associated with hypoaldosteronism [8]. The association between renal failure and greater increases in serum potassium levels may be explained physiologically: reduced GFR leads to decreased renal excretion of potassium, even within normal potassium levels [9,10,11]. Accordingly, the increase in serum potassium levels was greater following each subsequent supplementation of potassium dose, regardless of the dose. In contrast, the association between the use of mechanical ventilation and the greater increases in serum potassium levels cannot be explained by any known physiological effect of the ventilation itself. Since serum potassium levels were adjusted according to their pH scale, the direct possible effect of ventilation on PaCO2, and thus on the pH scale, was statistically nullified. However, similar to the use of vasopressors, we assume that the use of mechanical ventilation indicates sicker patients, who probably have higher cytokine levels.

A critical issue in replacing potassium losses is estimating the amount of supplement required. Our study shows that this amount is dependent on several measurable variables. Thus, modeling the response is possible, and may be used as a bedside tool to direct physicians in replacing potassium, while minimizing the need for frequent repeated measurements of serum potassium. Yet, relying on a prediction model while replacing potassium might be perilous, as unmeasured parameters, such as absorption of injected subcutaneous insulin or a sudden unnoticed change in a patient’s minute volume, may interfere with the model, and unexpectedly change its accuracy. Interestingly, the rise in serum potassium level was higher for any given accumulative potassium dose in patients with moderate than mild hypokalemia. We assume that a normal physiologic response to the degree of hypokalemia is responsible for this finding, e.g., potassium urine secretion decreases [12] as hypokalemia is more dominant, to enable rapid normalization of the serum potassium level. From the findings of this study, we generated a model to assist physicians in determining the personalized dose of intravenous potassium supplement, while minimizing the number of laboratory tests of potassium level. The data to be entered by the user are patients’ age, creatinine level, use of mechanical ventilation, use of vasopressors, and the given potassium dose. Ph is not needed to use this formula. This model is available (accessed on 10 March 2021) online: https://drive.google.com/file/d/1sJoQ83VodtlR31P2X4ii4JELxXqlAJdj/view?usp=sharing

Our study has a number of limitations. First, this is a single center study, and all blood tests were performed by our hospital laboratory services. Thus, a potential local laboratory bias may have affected the accuracy of the results. Second, to prevent blood from clotting in the tube, we routinely used heparin-coated syringes. It is known that heparin itself may cause a very mild increase in measured serum potassium levels [13]. Third, our study included only patients who were treated with potassium supplement via the intravenous route, and the only remedy that was used was potassium chloride. Additionally, we did not include patients with severe hypokalemia (i.e., potassium <2.5 mEq/L) due to their very low number, and we did not analyze the potential effect of lean tissue mass and muscle mass that could contribute to the increase in serum potassium levels in some patients.

## 5. Conclusions

We suggest that among critically ill patients with mild to moderate hypokalemia, the rise in serum potassium after intravenous potassium supplement is influenced by a wide spectrum of clinical parameters. Further, the variability in the yield between patients is high, even after adjustment for these parameters. We present a practical personalized formula to predict the expected rise in serum potassium based on relevant clinical parameters.

## Figures and Tables

**Figure 1 jcm-10-01986-f001:**
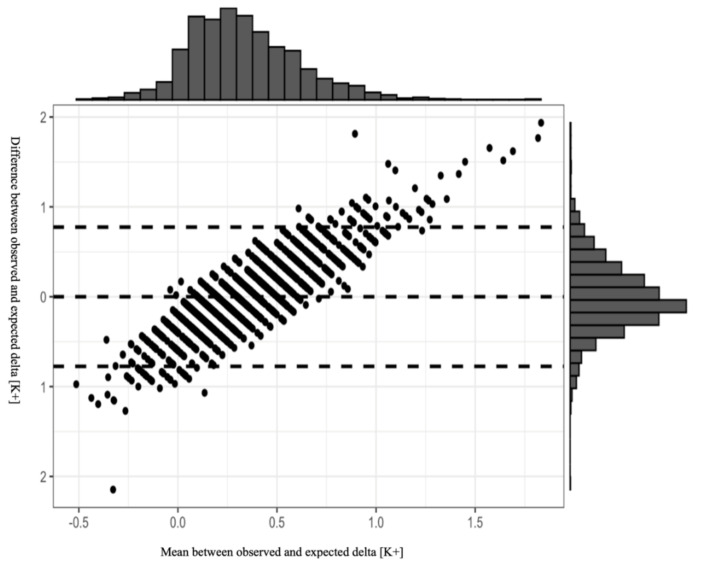
The results of the Bland–Altman plot of a model for estimating an effect of a given potassium dose. The model is based on the following parameters: patient’s age, serum creatinine, the need for mechanical ventilation, and the use of vasopressors. Most of the observations are within the 95% CI.

**Table 1 jcm-10-01986-t001:** Baseline characteristics of patients treated for hypokalemia.

	Total (*n* = 683)	Mild Hypokalemia (3–3.5 mEq/L) (*n* = 422)	Moderate Hypokalemia (2.5–2.99 mEq/L) (*n* = 261)	*p*- Value
Age	54.0 ± 21.4	54.5 ± 21.6	53.2 ± 21.2	0.448
Gender				
Male	54.3% (371)	52.1% (220)	57.9% (151)	
Diabetes mellitus	5.6% (38)	5.2% (22)	6.1% (16)	0.611
Hypertension	6.7% (46)	7.6% (32)	5.4% (14)	0.261
Ischemic heart disease	0.6% (4)	0.5% (2)	0.8% (2)	0.627
Cerebrovascular accident	2.3% (16)	2.4% (10)	2.3% (6)	0.953
Mortality in ICU	22% (155)	23.5% (99)	21.5% (56)	0.544

Data are presented as mean ± standard deviation or as % (*n*). ICU—intensive care unit.

**Table 2 jcm-10-01986-t002:** Characteristics of hospitalizations in the intensive care unit.

	Total (*n* = 683)	Mild Hypokalemia (3–3.5 mEq/L) (*n* = 422)	Moderate Hypokalemia (2.5- 2.99 mEq/L) (*n* = 261)	*p*- Value
Hospitalization days in the ICU	12.5 ± 14.2	11.9 ± 13.7	13.5 ± 15.0	0.152
Mechanical ventilation during ICU hospitalization	80.5% (550)	80.8% (341)	80.1% (209)	0.815
Mechanical ventilation duration (days)	6.95 ± 10.9	6.7 ± 11.0	7.4 ± 10.8	0.199
Vasopressors given during ICU hospitalization	12.2% (83)	11.1% (47)	13.8% (36)	0.302

Epinephrine (Adrenalin)-IV	0.4% (3)	0.2% (1)	0.8% (2)	0.309
Norepinephrine (Nor-adrenaline)-IV	10.8% (74)	10.0% (42)	12.3% (32)	0.346
Dopamine HCL-IV	0.9% (6)	0.9% (4)	0.8% (2)	0.805

Data are presented as mean ± standard deviation or as % (*n*). ICU—intensive care unit.

**Table 3 jcm-10-01986-t003:** Potassium lab results and treatment summary.

	Mild Hypokalemia (3–3.5 mEq/L) (*n* = 422)				Moderate Hypokalemia (2.5–2.99 mEq/L) (*n* = 261)			
	Potassium Intravenous Dose				Potassium Intravenous Dose			
	Dose of 20 mEq (*n* = 96) (22.8%)	Dose of 40 mEq (*n* = 277) (65.6%)	Dose of 60 mEq (*n* = 8) (1.9%)	Dose of 80 mEq (*n* = 41) (9.7%)	Dose of 20 mEq (*n* = 29) (11%)	Dose of 40 mEq (*n* = 194) (74.2%)	Dose of 60 mEq (*n* = 9) (3.5%)	Dose of 80 mEq (*n* = 29) (11.3%)
Potassium levels before treatment	3.2 ± 0.2	3.2 ± 0.2	3.2 ± 0.2	3.1 ± 0.2	2.6 ± 0.5	2.7 ± 0.5	2.5 ± 0.6	2.6 ± 0.5
Potassium levels after treatment	3.4 ± 0.4	3.5 ± 0.4	3.6 ± 0.4	3.4 ± 0.5	3.0 ± 0.6	3.2 ± 0.5	3.1 ± 0.7	3.0 ± 0.6
Change in potassium levels *	0.2 ± 0.4	0.3 ± 0.4	0.4 ± 0.4	0.3 ± 0.4	0.4 ± 0.6	0.5 ± 0.5	0.5 ± 0.5	0.4 ± 0.6
Change in potassium levels coefficient of variation (CV)	2%	1.3%	1%	1.3%	1.5%	1%	1%	0.6%

* A trend test for the change in potassium levels was performed for each group. *p*-values are 0.012 and 0.420, for mild and moderate hypokalemia, respectively. Means ± standard deviations (SD) are presented.

**Table 4 jcm-10-01986-t004:** Linear mixed model results for increases in serum potassium levels.

	Delta in Serum Potassium Levels			
Predictors	Estimates	CI 95%		*p*-Value
		Min	Max	
(Intercept)	0.22	0.11	0.32	<0.001
IV potassium—Dose of 40 mEq *	0.15	0.09	0.20	<0.001
IV potassium—Dose of 60 mEq *	0.16	0.02	0.30	0.030
IV potassium—Dose of 80 mEq *	0.16	0.07	0.25	<0.001
Creatinine for each increase of 0.1 mg/dL	0.03	−0.01	0.07	0.165
Mechanical ventilation	0.02	−0.05	0.09	0.643
Vasopressor	0.08	−0.01	0.16	0.080
Age	0.00	0.00	0.00	0.093
Moderate hypokalemia **	0.20	0.15	0.26	<0.001

* Reference group—IV potassium—Dose of 20 mEq; ** Reference group—Mild hypokalemia (3–3.5 mEq/L); IV—intravenous; CI—confidence interval.

## Data Availability

The data used in the analysis of this study are not publicly available due to national regulations but are available from the corresponding author upon request, and following the Ethics Committee’s approval.

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
