# Peer review of "Estimation of Potassium Changes Following Potassium Supplements in Hypokalemic Critically Ill Adult Patients–A Patient Personalized Practical Treatment Formula"

_jcm, 2021, doi:10.3390/jcm10091986_

Round 1

Reviewer 1 Report

Thank you authors for conducting this research and allowing me to provide you the review. My comments are as followed:

  • The objectives of this study are interesting and I believe that the findings are highly probable to impact the clinical practice
  • Table 3, please add the number of sample size for each increment of potassium dose columns. These numbers are available in Table 4 and I think mentioning the sample size in Table 3 would be convenient to the audience
  • In Table 4, what does the number "1088" and "635" represent?
  • The overall manuscript is well written. However, some of the contents are missing in the Discussion. The authors reported that the potassium changes were associated with creatinine level, gender, mechanical ventilation, and vasopressors. However, I only saw a paragraph about vasopressors. Please discuss why Cr, gender, and mechanical ventilation are associated with the potassium changes. This could be in a separate paragraph in Discussion.
  • The Limitations are too short. Please make Limitations as a separate paragraph and explain more on each limitations you mentioned. For example, you stated that this study is single centered. Explain why this limits the applicability or generalization of your findings.
  • Some limitations were not mentioned. For example, the reported potassium levels from clotted blood tube and heparin tube could result in slight difference in potassium level. This factor was not controlled in your study. 

Reviewer 2 Report

The authors undertook a review of 683 critically ill patients classified as having either mild or moderate hypokalaemia upon admission, determining the impact of potassium supplements on the rise in serum potassium concentration ([K+]). They determined that patients with mild hypokalaemia responded to cumulative dose K+ supplement of between 20 – 80 mEq with increases in serum [K+] of between 0.2 – 0.4 mEq/L and for patients with moderate hypokalaemia of between 0.4-0.5 mEq/L. Larger increases were found in moderate hypoK. There was considerable variability in patient [K+] responses and a prediction model was developed to characterize these increases, based on patient demographics and broad clinical information. Whilst the idea is not novel, this is not a concern due to the common clinical practice of K+ supplementation and its very high clinical importance.

Statistical significance presentation in the ms was sloppy – surely the authors cannot be saying p<0.5 is acceptable? Authors present different components of the model as being significant yet their statistics indicate these were not. Presentation of which findings were different to others was very unclear.

It is not clear why [K+] was corrected for pH. In vivo pH clearly helps modulate [K+] but since the final variable is [K+] then this is not necessary. Were all [K+] corrected? Or was this just for the mixed model estimates? Nowhere apart from first mention is K+ corr mentioned – this is very confusing.

Was patient body mass measured or estimated in this cohort? Lean tissue mass - primarily representing skeletal muscle is likely to be a major factor contributing to the increases in [K+] observed. Were some patients cachectic? Age range is large which might argue against such an effect due to decrease in lean tissue expected beyond 50 years of age but this should be at least considered and discussed.

Title.

The title refers to K+ deficit, but the deficit per se is never measured, all that is measured is in fact the serum [K+]. Revise.

Abstract.

The authors refer to potassium “level” throughout the ms when in fact they should be more precise and refer to “concentration”. The sentence on cumulative doses in abstract is unclear and should be revised to clarify is absolute cumulative dose.

  1.  

Skeletal muscle mass is a major factor affecting K+ regulation due to its very high fraction of body mass (30-50%), high content of Na+,K+-ATPase, regular blood flow, adaptability and large capacity to take up K+ (see numerous reviews by Torben Clausen, as well as a major store of K+ to release K+ and thus minimize hypokalaemia (see numerous reviews by Alicia McDonough) and this was not considered in the introduction, methods, results or discussion. Were any intramuscular proteins measured in these cohorts eg myoglobin, contractile proteins that might give a crude gauge of muscle deterioration?

The length of K+ depletion / K+ clearance problems impacts upon development of hypokalaemia. Was this considered in the analyses?

If intent is to have clinicians use the prediction equation do they first need to correct [K+] for pH? This is very unclear. It would be important to present the pH data also rather than correct for it.

Ventilation was assessed for contribution to predicted [K+] and reported to be a factor. Introduction mentions this effect is likely due to alkalosis but this is not discussed. Were patients in fact exhibiting respiratory alkalosis? Again presentation of pH would be informative.

  1.  

Were patients body mass or lean tissue mass assessed?

Where was blood sampled from? Was it venous or arterial blood? Was blood collected with stasis for venous blood or with muscle contractions of the forearm? These can have a major impact on the actual [K+].

K+ correction – see above. When were “initial” and “final” pH measured? Was this before and after the K+ supplementation? Very unclear. It is also pH not PH. In absolute terms [H+] correction would be better than correction of a logarithmic eg delta pH 7.4 to 7.2 is vastly different to 7.2 to 7.0 in terms of changes in [H+]. Is it really needed?

Include that patients with severe hypokalaemia were excluded.

Hypokalaemia was defined as 2.5 – 3.5 mEq/L, it would be more accurate to define as < 3.5 mEq/L? For example 2.0 mEq/L is definitely hypokalaemia!

  1. Results

3.1 Study Population

Tables 1 and 2 should be standardized with placement of “Total”, suggest in first column as then it is clearer that statistical comparison is between mild and moderate hypokalaemia.

Table 1. Mortality is expressed as n (%) whereas rest of table is % (n) - please standardize

3.2 Characteristics

This section needs to be rewritten. It describes a number of “differences” that were not significantly different, therefore were not differences.

What vasopressors were used? This may enable interpretation of their effect on [K+]. Give exact p value here in comparisons between cohorts.

3.3 Serum [K+]

Description of statistics for each rise in [K+] was not provided, rather one stat level for each clinical cohort. This needs clarification. The text duplicates the table numbers.

Table 3 – it is not clear what comparison the p values actually refer to?

Did the change in [K+] differ significantly between cumulative doses? If not authors cannot say proportional increases…

“Change in” [K+] seems imprecise – I presume authors mean increase? Or were decreases actually seen in some patients?

Move Mean ± SD to bottom of table

Unclear what “coeffect of variance” refers to

Table 4. this can be merged into Table 3. Why does total “n” at bottom of table differ from total number of patients? Does this mean some patients received multiple supplements on a given hospitalisation? Or were they repeat hospitalisations? Unclear

3.4 Linear mixed model results

Data is presented in Table 5 showing that none of the creatinine (p=0.165), mechanical ventilation (p=0.643) vasopressor use (p=0.08, tendency) or age (p=0.092, tendency) statistically affected the [K+] prediction. From these statistical findings there does not seem to be support to include these in the model. Also the discussion on this seems unarranted in that context. Table indicates “blood potassium” - it is not blood potassium which includes red blood cells with much higher [K+], it is serum [K+]

Equation should include brackets for clarity

Figure 1 “mean of potassium levels” is incorrect. Isn’t this the delta[K+]? This shows the model becomes unreliable at high delta[K+]?

Other

What were pH of the patients upon admission and during supplementation?

  1.  

Specify which vasopressors were commonly used. This may be important in interpretation – eg could this effect be due to impacts on tissue perfusion of eg skeletal muscle/kidney/liver? Some of these may also directly stimulate or inhibit the Na+K+ATPase.

Explain the link with ventilation.

Muscle mass might also be a correlate of patient sickness.

It is very interesting that the response to a given cumulative dose in moderate hypoK was higher in moderate compared to mild hypoK, but this was not discussed?
